# Association of Nutrition in Early Childhood with Body Composition and Leptin in Later Childhood and Early Adulthood

**DOI:** 10.3390/nu13093264

**Published:** 2021-09-18

**Authors:** Louise R. Jones, Pauline M. Emmett, Nicholas P. Hays, Yassaman Shahkhalili, Caroline M. Taylor

**Affiliations:** 1Centre for Academic Child Health, Bristol Medical School, University of Bristol, Bristol BS8 1NU, UK; p.m.emmett@bristol.ac.uk (P.M.E.); caroline.m.taylor@bristol.ac.uk (C.M.T.); 2NPTC Nutrition—SBU Nutrition, Avenue Nestle 55, 1800 Vevey, Switzerland; NicholasPaul.Hays@nestle.com; 3Nestlé Institute of Health Sciences, 1000 Lausanne, Switzerland; yasushahkhalili@yahoo.fr

**Keywords:** ALSPAC, energy, dietary fat, body composition, obesity, leptin

## Abstract

Objectives: Using data from the Avon Longitudinal Study of Parents and Children (ALSPAC), this study aimed to replicate the finding of the Etude Longitudinale Alimentation Nutrition Croissance des Enfants (ELANCE) that low fat intake in early childhood was associated with increased adiposity in adulthood. Methods: Diet was assessed at 8 and 18 months using 3-day food records. Body composition variables were measured at 9 and 17 years, and serum leptin at 9 years. Associations were modelled using adjusted linear regression. Results: In replication analyses, in contrast to ELANCE, there was a positive association between fat intake (% energy) at 18 months and fat mass (FM) at 9 years (B coefficient 0.10 (95% CI 0.03, 0.20) kg, *p* = 0.005). There was no association with serum leptin. In extended analyses fat intake at 18 months was positively associated with FM in boys (0.2 (0.00, 0.30), *p* = 0.008) at 9 years but not in girls. Fat intake was positively associated with serum leptin concentration in boys (0.2 (0.1, 0.4) ng/mL, *p* = 0.011) but not in girls. Conclusions: Our results did not corroborate the findings from the ELANCE study. A high fat diet in early life may have implications for later childhood and adolescent obesity.

## 1. Introduction

Childhood obesity is a major public health issue and its prevalence is increasing both in the UK and internationally. The World Obesity Federation Atlas published in 2019 forecast that childhood obesity will rise markedly by 2030 [1]. To develop strategies for the prevention of obesity and thereby avoid long-term health effects [2,3,4,5,6], it is important to identify modifiable predisposing factors.

Obesity research has focused on nutrition in infancy and early childhood and associations between early life influences and adult cardiovascular and metabolic diseases have become apparent [7]. Studies investigated the influence of early nutrition on later obesity risk but there is no agreement about the roles of specific macronutrients [8]. In early childhood a high protein intake has been linked to later body composition as it enhances secretion of insulin-like growth factor-1, which may increase growth and adiposity [9]. Dietary fat intake is reduced during complementary feeding with the partial replacement of milk by carbohydrate-rich foods [10,11]. During this transition period many important hormonal and enzymatic changes occur that affect carbohydrate and lipid metabolism [12]. A high carbohydrate, low fat diet at this early age has been shown to increase insulin and decrease plasma glucagon concentrations in infants of different species, including humans [12]. Animal studies have shown that a high carbohydrate, low fat diet during both the suckling [13] and weaning period [14] programs susceptibility to obesity later in life. The programming potential of the fat content of early diet on the development of adiposity in humans is not known.

A few longitudinal studies have investigated macronutrient intake and body composition using a range of outcome measures including BMI, skinfold thicknesses, and fat and lean mass, but with differing results. [8,15]. The ELANCE study followed a small group of children longitudinally for 20 years with outcomes at 8 years (*n* = 112) [16] and 20 years of age (*n* = 73) [17]. At 8 years a positive association between protein intake at age 2 years and body fatness was found but no relationship with fat intake [16]. At 20 years the study suggested that low fat intake in early childhood increased the susceptibility to the development of obesity and leptin resistance in adulthood, and that early programming of leptin resistance might be one way in which nutrition in infancy could affect adiposity development [17].

The ELANCE participants had low intakes of dietary fat for this age: the mean fat %E at 10 months was 27.7% (SD 4.8%) for boys and 28.2% (SD 4.4%) in girls; in comparison at 8 months of age the ALSPAC participants had a mean fat %E intake of 35.2% (SD 5.0%) for boys and 35.6% (SD 5.0%) for girls. WHO recommends 30–40% fat %E during weaning until 3 years [18]. There is a strong case for repeating these analyses in another setting where fat intakes are more varied, with a larger cohort size to increase the investigative power and additional confounders that have associations with childhood obesity [19,20,21,22]. Furthermore it is relevant to analyse boys and girls separately since the *Atlas of Childhood Obesity* reported a large difference in the prevalence of obesity by sex, with a greater prevalence among boys [1]. These sex differences may be due to biological differences in body composition or hormonal differences [23].

The Avon Longitudinal Study of Parents and Children (ALSPAC) collected diet data at similar timepoints to the ELANCE cohort (8 and 18 months of age) and includes body composition data at similar ages (9 and 17 years) and serum leptin (9 years). The aims were therefore: (1) to investigate the generalisability of the findings in ELANCE by replicating the investigation in ALSPAC and (2) to extend the investigation with additional analyses to take advantage of the larger sample size and range of confounders available.

## 2. Methods

### 2.1. Study Sample

ALSPAC is a geographically based prospective cohort study investigating factors influencing health, growth and development. Pregnant women resident in Avon, with expected dates of delivery 1 April 1991 to 31 December 1992 were invited to join. Initially there were 14,541 pregnancies enrolled, with 14,062 live births and 13,988 children alive at 1 year of age [24,25]. A proportion of children born in the last 6 months of the study (*n* = 1432) were recruited to a sub-study called Children in Focus (CIF). Research clinics were held between 4 and 61 months of age. Ethics approval for the study was obtained from the ALSPAC Ethics and Law Committee and the Local Research Ethics Committees in writing. The study website contains details of all data that are available through a fully searchable data dictionary and variable search tool at http://www.bristol.ac.uk/alspac/researchers/our-data/ (accessed 16 September 2021).

### 2.2. Dietary Assessment

Dietary information was collected using three 1-day food records when the participants were aged 8 and 18 months. Parents were asked to record everything the child consumed, using household measures and to include two weekdays and one weekend day. A fieldworker checked the records with the carer to clarify any anomalies. The methodology has been described in detail elsewhere [26,27]. Mean daily energy and macronutrient intakes were calculated. All macronutrients were expressed as their percentage contribution to energy. Total energy was expressed in increments of 0.42 MJ, equivalent to 100 kcal.

### 2.3. Misreporting

The degree of misreporting of energy was assessed using the Davies method [28] for pre-school children. This compares predicted energy intake (PEI) with reported energy intake (REI). PEI is body weight (kg) × energy requirement (MJ/kg). Energy requirement was defined as 0.4 MJ/kg, the estimated average requirement for infants aged 6–36 months [29]. Children were categorised based on the difference between PEI and REI, allowing a tolerance of approximately 22% of the mean energy intake for that age group.

### 2.4. Child Anthropometry

At 9 and 17 year clinics height was measured using a Harpenden Stadiometer and weight was measured to the nearest 50 g using a Tanita Body Fat analyser (model TBF 305). Whole-body DXA scans, using a Prodigy scanner were carried out to derive total fat mass (FM) and fat-free mass (FFM).

### 2.5. Serum Leptin

Non-fasting blood samples were collected using a standard procedure during clinics held during the day at 9 years. The samples were centrifuged and frozen at −80 °C. Serum leptin was measured by an in-house enzyme linked immunosorbent assay (ELISA) validated against commercial methods.

### 2.6. Confounding Variables

Maternal smoking in pregnancy was derived from self-completion questionnaires completed at 18 and 32-weeks’ gestation. Maternal educational level collected at 32-weeks’ gestation was categorised as: Low (none, Certificate of School Education (CSE), vocational and Ordinary (O) level) and high (Advanced (A) level or degree). CSE and O levels were assessed at 16 years of age; A levels at 18 years of age. Paternal social class was collected from the same questionnaire: it was based on the registrar general’s classification of occupations; fathers were classified as unskilled/semiskilled vs. skilled/professional. Breastfeeding duration was collected by questionnaire when the infants were 6 months old. In the ELANCE replication models it was categorised as never breastfed vs. any breastfeeding. In the ALSPAC models breastfeeding was categorised as never breastfed <1 month vs. ≥1 month. Pre-pregnancy height and weight were self-reported by parents and used to calculate their BMI (weight/height^2^).

### 2.7. Statistical Analysis

The associations between dietary intakes at 8 and 18 months and body composition at 9 and 17 years and serum leptin concentrations at 9 years were analysed using adjusted linear regression models. Goodness of fit was assessed by adjusted R^2^ values and standard errors of the estimates. All statistical analyses were performed using the SPSS for Windows statistical software package version 24 (SPSS Inc., Chicago, IL, USA).

*Replication analyses*: The analyses were performed with sexes combined and adjusted for, birthweight, breastfeeding, maternal BMI and paternal social class to replicate the ELANCE analyses.

*Extended analyses*: Extended analyses were performed to make use of the range of data available in ALSPAC. Two models were investigated. Confounders for model 1 were maternal smoking in pregnancy, maternal education, birthweight, maternal BMI and breastfeeding; in model 2 Paternal BMI was added, this was due to less data being available on fathers with a consequent reduction of power. For analysis with FFM and FM, height was added to adjust for body size. Regressions were performed separately for each sex with all participants and again with plausible reporters only. These analyses were repeated after multiple imputation for missing data. Multiple imputation by chained equations was used to impute missing data and 25 datasets were generated. All outcome variables and all variables included in the adjusted analyses were used to impute the data. In models with macronutrients, energy was added as a confounder in accordance with the multivariate nutrient density model [30]. For further analysis the children aged 18 months were categorised into three groups using fat %E based on an international reference value for fat intake for that age group of 30–40 %E (group 1 < 35 fat %E; group 2 35–40 fat %E; group 3 > 40 fat %E) [18,31]. There were too few children with fat %E < 30% to analyse separately. Regression analyses were performed to investigate associations between these groups and body composition at 9 and 17 years.

## 3. Results

### 3.1. Characteristics of ALSPAC-CIF Children

Of the 1432 children recruited to the CIF study, 1177 had dietary data at 8 months and 1025 at 18 months. The baseline characteristics of participants who had body composition data at 9 and 17 years and those without are shown in Appendix A. There were no differences between the groups for birthweight, birth length or BMI at 8 and 18 months (all *p* > 0.05). Energy intakes were higher at 8 and 18 months for those lost to follow up (*p* = 0.037, *p* < 0.0001, respectively) with no differences for other macronutrients. Those lost to follow up were more likely to be boys, have younger mothers, with lower educational attainment, and less likely to be breastfed (all *p* < 0.001).

Nutrient intakes at 8 and 18 months stratified by sex are presented in Table 1. At 8 months 22.6% were breastfed either exclusively or in combination with formula, 64.1% had formula milk with or without cows’ milk, and the remainder had cows’ milk (mostly full fat) as their milk drink. All were having solid foods. By 18 months most were consuming cows’ milk only: 75.1% full fat and 6.4% low fat milk. A few were consuming breast milk (2.3%) or formula milk (7.2%). At 9 years the only difference between boys and girls was the proportion of FM and FFM. At age 17 boys were taller and heavier but the girls had greater FM. Girls had significantly greater mean leptin concentration.

### 3.2. ELANCE Replication Analyses

The analyses replicating ELANCE are presented in Appendix A (diet at 8 months) and Appendix A (diet at 18 months). Very few associations were present at 8 months: there was a positive association between energy intake and BMI at 9 years after adjustment (*p* = 0.044). A similar association was found with energy intake at 18 months (*p* = 0.033) which was associated with FFM but not FM and not present at 17 years. Monounsaturated fatty acid intake (MUFA) %E was positively associated with BMI at both 9 (*p* = 0.022) and 17 (*p* = 0.044) years in the adjusted models. At 18 months strong positive relationships were found between fat %E (*p* = 0.005) and MUFA %E (*p* = 0.001) and FM at 9 years, and there was a positive trend between MUFA %E and FM at 17 years (*p* = 0.055). There was no association of energy or any macronutrient with serum leptin concentration.

### 3.3. Extended Analyses

Using diet at 8 months of age (Appendix A) for all participants there were few associations in either sex between energy intake or %E from any macronutrients and later body composition or leptin concentration at age 9 years after adjustment. In boys only, there was weak evidence of a negative relationship between total fat %E and FFM at age 17 years (*p* = 0.035 model 2). No robust associations were present after restriction to plausible reporters, or multiple imputation (data not shown).

Using diet assessed at 18 months of age for boys (Table 2 and Table 3) there were positive associations in model 2 between energy intake (all participants) and BMI and FFM at 9 years but not at 17 years. When restricting to plausible reporters these associations with energy intake were much stronger at 9 years (both *p* < 0.001) and a positive association was also present for FFM at 17 years (*p* = 0.015). There were positive associations between fat %E, SFA %E, MUFA %E and FM at 9 years, for all participants and for plausible reporters, with reverse associations for carbohydrate %E. In model 2, a 1% increase of energy from MUFA was associated with a 0.4 (95% CI 0.1, 0.7) kg increase in FM in boys at 9 years. The associations of fat %E and MUFA %E with FM were present again at 17 years. There were negative associations between %E from total fat, SFA, MUFA and FFM at 9 and 17 years in all participants. On restriction to plausible reporters the associations with FFM were not present. The results were similar following multiple imputation (data not shown). There were positive associations of total fat, SFA and MUFA E% with leptin concentrations at 9 years in model 2 in all participants that were attenuated in plausible reporters. A negative association of carbohydrate %E was present in all participants but not when restricted to plausible reporters. Protein %E was not associated with any outcomes.

When diet at 18 months (Table 4 and Table 5) was analysed for all girls there were no associations between energy or any macronutrients with body composition or leptin at age 9 years. However, in the plausible reporters, associations with energy intake were evident: positive associations with BMI and FM at 9 years, in the fully adjusted model. An increase in energy intake of 0.42 MJ (100 kcal) was associated with a 0.5 (95% CI 0.1, 0.9) kg increase in FM. There were no associations with energy intake at 17 years. At 17 years in all girls there were associations of some nutrients with FFM, positive with carbohydrate %E (*p* = 0.018) and negative with SFA %E (*p* = 0.012), which were not present for plausible reporters only.

When the children’ diets were categorised at 18 months in relation to the reference intake for fat %E, we found differing results between boys and girls (see Table 6). The boys with intakes <35%E from fat had lower mean BMI and FM at 9 and 17 years than those with intakes >40%; this was a linear relationship. There was no association with FFM (data not shown). In girls there was no association with FM at either age. The association with BMI at 17 years was not linear.

Adjusted R^2^ values and SEE of the extended models are shown in Appendix A.

## 4. Discussion

Our results suggest that a diet high in fat and low in carbohydrate in early childhood is associated with increased adiposity later in life in boys in this cohort. There was strong evidence among boys of a positive association between fat %E at 18 months and FM at age 9 years (0.2 (95% CI 0.0, 0.3) kg, *p* = 0.008) and the association was present but attenuated at age 17 years (0.3 (95% CI 0.0, 0.6) kg, *p* = 0.045). This equated to a difference of 2.3 kg (at 9 years, *p* = 0.003) and 5.1 kg (at 17 years, *p* = 0.009) in FM between boys with fat intakes <35% compared with intakes >40% of energy (see Table 6), a substantial effect. This was strongly associated with the MUFA %Fat. There was also evidence among boys, particularly in plausible reporters, of a positive association between energy intake at 18 months and BMI at 9 years with the increase being in FFM rather than FM, evident at both 9 (0.3 (95% CI 0.2, 0.5) kg *p* ≤ 0.001) and 17 (0.7 (95% CI 0.2, 1.3) kg, *p* = 0.015) years. In girls there were few associations which were inconsistent.

In ELANCE at 8 years of age they found a positive relationship between energy intake in early childhood and BMI [16]. In our replication we found similar associations with BMI at 9 years, although the association was with FFM not FM so is a partial replication. In contrast to ELANCE, our study did not support the hypothesis that early protein intake is associated with obesity in mid childhood, although protein intakes were lower in ALSPAC. In ALSPAC there was evidence of a positive association between fat intake at 18 months and fat mass at 9 years, this was not evident in the ELANCE study.

In the second ELANCE study at 20 years of age [17] the authors found no relationship with protein intake, but there were negative relationships between total fat %E at 2 years and FM and subscapular skinfold thickness, which was related to SFA, leading to the conclusion that low fat intakes in the first 2 years of life were associated with increased body fat in later years [16]. Our study found fewer associations at 17 years but did confirm a positive association between energy intake and FFM at least in boys. In contrast to ELANCE we found the opposite association between total fat, SFA and MUFA %E and FM: as fat intake increased so did FM. Our findings do not support the conclusions of the ELANCE study regarding fat intake.

There are several differences between ELANCE and ALSPAC that could account for the conflicting results. ELANCE had a much smaller study sample: 112 at age 8 years and 73 at age 20 years compared with >700 at 8 years and >400 at 17 years in ALSPAC. Fat intakes in ELANCE were particularly low: at 10 months mean fat %E was 28% and at 2 years 32%. These were below the mean intakes reported in ALSPAC of 35% at 8 months and 37% at 18 months and below international reference intakes of 40–60% during infancy gradually reducing to 30–40% by 24–36 months [18,31]. In ELANCE instead of fat %E intake reducing as the participants got older, intake increased to 38% at 8 years of age. It was proposed that this change from an early low to a later high fat intake may promote the development of adiposity [32]. In ALSPAC fat intake stayed close to the reference intakes in early years [26,27] and remained stable throughout childhood [33,34]. The low fat intakes in ELANCE could be explained by their milk intake: at 10 months no children were breastfed, 44% consumed formula milk and the remainder were having cows’ milk mostly with reduced fat levels.

In our study there was a relative increase in fat intake between 8 months and 18 months and this was all in saturated fat fraction and reflected the move from infant foods to family foods. The types of foods which were eaten more frequently by children at 18 months than at 8 months were those containing fat and sugar; for example, the proportion of children eating biscuits rose from 35% to 87% and those eating cakes/puddings rose from 38% to 58% [26,27]. This may account for the fact that we only found associations with the diet at 18 months. Our results support recommendations for feeding young children that suggest limiting these fat and sugar containing foods [35].

Other studies have examined early diet in relation to body composition pre- and post-adolescence. These studies have used different measures of body fatness both direct and indirect but have reported similar results to our study. Fat intakes recorded between 2 and 8 years were positive predictors of BMI at 8 years [36] with an inverse association for carbohydrate, while in 9–11-year-olds [37] percentage of body fat correlated positively with fat intakes. Across the age range 2–15 years energy—adjusted fat intakes were positively, and carbohydrate intakes inversely related to greater skinfold thicknesses [10]. In our study adiposity was measured directly by DXA, so we were able to explore the positive relationship between fat intake and BMI and show that it is driving a greater change in FM than FFM in boys as reported previously [38]. Our findings also showed that higher energy intake in boys at 18 months was associated with a greater increase in BMI specifically FFM at 9 years but in girls increased energy intake resulted in higher FM. Other studies have corroborated that as BMI increases throughout childhood boys lay down more FFM whereas girls accumulate more FM [39,40].

We were able to study boys and girls separately which is important as there are biological differences in body composition between the sexes. We found many associations between diet and body composition in boys but less in girls. Few studies have looked at boys and girls separately, but two found an association between dietary fat and FM in boys but not girls similar to our study [41,42]. There were also differences observed for serum leptin levels between boys and girls. As fat intake increased for boys there was a concurrent increase in FM and leptin. This is unremarkable as leptin is a hormone that is produced by fat cells and is positively correlated with the amount of body fat present [39,43]. The variations between the sexes may be due to biological differences: girls naturally have higher FM and circulating serum leptin concentrations than boys [44], so it is possible that this masks any association between dietary fat %E and body composition and leptin. In addition, the variation in fat %E in this study may not be large enough to show an effect in girls.

In our study we found a strong positive association between the fatty acid types MUFA %E and SFA %E and FM corroborated in another study in which percentage of body fat in children aged 9–11 years correlated positively with intakes of all types of fat and negatively with carbohydrate [37]. This complementary relationship between carbohydrate and fat (as fat %E increases it displaces carbohydrate %E) was also present in our study. We did not find evidence that a diet low in fat and high in carbohydrate in early childhood programmes adiposity in later life, and our findings in boys were the reverse of this. However, a study that investigated children’s food and nutrient intake including MUFA in pre-school years and body composition in mid childhood found that higher longitudinal intakes of dairy products and MUFA were associated with lower body fat determined by DXA [45] in contrast to our findings.

In boys but not girls, intakes of fat <35%E in early childhood were associated with lower BMI and FM in later childhood and young adulthood compared with intakes of fat >40%E, thus validating, at least in boys, the recommendations for a fat intake not to exceed 40%E. Previously in ALSPAC we have shown that at 18 months full fat dairy products are a primary source of fat and fatty acids in the diet [46]. Children with higher fat diets consumed more whole milk, and cheese. Some children with the highest fat intakes were consuming in excess of 750mL milk per day. To reduce fat intake children’s whole milk intake could be limited to 500mL or changed to semi-skimmed milk from the second year of life provided they were eating a well-balanced diet [46].

Our study has many strengths including: (1) the long follow up period from early childhood to early adulthood; (2) a large sample size; (3) repeated measures of diet and the ability to identify plausible reporters; (4) anthropometric measures using DXA rather than skinfold thicknesses or bioimpedance; (5) a wide range of confounding variables available for analyses, including maternal smoking in pregnancy which has been associated with childhood obesity [19,20], maternal education which has been identified as a good proxy for social class and is associated with childhood obesity in ALSPAC [21], and paternal overweight\obesity as well as maternal overweight\obesity which are associated with obesity in offspring [22]. As we did not have complete data for fathers available, to retain full power in our main model we ran a second model to include the fathers’ data.

There are also limitations. (1) The associations could be different among participants and non-participants, and the study was undertaken in one geographical area of the UK, limiting its generalisability. (2) Participants were lost to follow up, reducing the amount of data available at the later time points; multiple imputation showed that this was unlikely to have biased our findings. (3) It is difficult to assess dietary intakes accurately, but this was addressed by measuring the level of misreporting. Restricting to plausible reporters attenuated some results but strengthened others: the main findings for the association between fat %E and FM and energy and FFM in boys remained. Reporting over a relatively short period of 3 days may also not be fully representative of habitual diet in children beyond the preschool years [47]. (4) Attrition in a longitudinal cohort is inevitable and is likely to result in the cohort over-representing participants of greater socioeconomic status. This can potentially cause bias and limit generalisability [48]. (5) There may be other confounders that we were unable to account for in the extended analysis. We did not include adjustment for adult nutrition in the replication of the ELANCE study, to mirror their adjustment. We were unable to include adjustment for adult nutrition in the extended analyses as these data were not available in ALSPAC. The samples for leptin were obtained during the day, but we do not have any data on the exact time of day for each sample, so that we were unable to account for diurnal variation. (6) The results of the goodness of fit tests showed greater adjusted R^2^ values in some cases than we would expect for similar studies. Prediction intervals calculated from SEE were generally acceptable (>90%) especially at 7 years old. Lower values in some instances may reflect relatively low case numbers in each category and highlights the importance of replication of this study in other cohorts.

In conclusion our results did not corroborate the findings from the ELANCE study. We found that as fat intake in early childhood increased so did FM in late childhood and early adulthood. Energy intake at 18 months was positively associated with FFM in later life. However, these findings were only apparent in boys. Dietary composition in early life may have implications for later childhood and adolescent obesity: this needs further investigation in other cohorts with longitudinal data.

## Figures and Tables

**Table 1 nutrients-13-03264-t001:** Descriptive statistics for nutrient intakes at 8 months and 18 months and body composition at birth, 9 and 17 years in boys and girls enrolled in Avon Longitudinal Study of Parents and Children (ALSPAC)-Children in Focus (CiF).

	Boys	Girls	*p*-Value
Dietary intakes			
*8 months*	*n* = 642	*n* = 534	
Energy, MJ	3.51 (0.73)	3.28 (0.71)	<0.0001
Protein, %E	13.8 (2.7)	13.6 (2.7)	0.277
Carbohydrate, %E	50.9 (5.7)	50.6 (5.7)	0.370
Fat, %E	35.2 (5.0)	35.6 (5.0)	0.141
Saturated fat, %E	15.4 (3.5)	15.7 (3.4)	0.221
Monounsaturated fat, %E	12.4 (2.1)	12.6 (2.2)	0.124
Polyunsaturated fat, %E	5.0 (1.5)	5.0 (1.5)	0.791
*18 months*	*n* = 562	*n* = 463	
Energy, MJ	4.75 (0.94)	4.43 (0.90)	<0.001
Protein, %E	15.2 (2.3)	15.4 (2.3)	0.401
Carbohydrate, %E	47.5 (5.6)	47.3 (5.6)	0.607
Fat, %E	37.4 (4.7)	37.4 (4.8)	0.826
Saturated fat, %E	17.9 (3.5)	18.0 (3.6)	0.688
Monounsaturated fat, %E	12.0 (1.8)	12.0 (1.7)	0.903
Polyunsaturated fat, %E	4.3 (1.5)	4.2 (1.5)	0.385
Anthropometry			
*Birth*	*n* = 766	*n* = 651	
Birth weight, kg	3.5 (5.5)	3.4 (4.9)	<0.0001
Length at birth, cm	50.9 (2.2)	50.2 (2.0)	<0.0001
*9 years*	*n* = 483	*n* = 433	
Weight, kg	34.5 (7.3)	34.9 (7.5)	0.143
Height, cm	139.5 (5.8)	138.9 (6.1)	0.766
BMI, kg/m^2^	17.7 (2.8)	17.9 (3.0)	0.138
Fat-free mass, kg	25.4 (2.9)	23.5 (3.1)	<0.001
Fat mass, kg	7.6 (5.0)	9.6 (5.0)	<0.001
*17 years*	*n* = 289	*n* = 327	
Weight, kg	72.7 (14.4)	62.4 (12.5)	<0.001
Height, m	178.4 (5.9)	165.0 (6.1)	<0.001
BMI, kg/m^2^	22.8 (4.5)	22.9 (4.4)	0.774
Fat-free mass, kg	54.7 (6.5)	37.9 (4.3)	<0.001
Fat mass, kg	14.1 (10.3)	21.5 (9.4)	<0.001
Hormonal status	*n* = 310	*n* = 282	
Serum leptin (ng/mL)	6.9	10.1	<0.0001

Values are mean (SD); %E, percentage of energy.

**Table 2 nutrients-13-03264-t002:** Multiple linear regression models for energy and energy-adjusted macronutrient intakes of boys, using diet collected at 18 months of age to predict body composition and serum leptin concentration at 9 years of age and body composition at 17 years of age in children enrolled in ALSPAC-CiF.

	Model	Energy/0.42 MJ		Protein (%E) ^a^		Carbohydrate (%E) ^a^		Fat (%E)					
								Total ^a^		SFA ^a^		MUFA ^a^	
		B (95% CI)	*p*-Value	B (95% CI)	*p*-Value	B (95% CI)	*p*-Value	B (95% CI)	*p*-Value	B (95% CI)	*p*-Value	B (95% CI)	*p*-Value
9 years													
BMI, kg/m^2^	Model 1 ^b^ (*n* = 360)	0.2 (0.0, 0.3)	0.009	0.0 (−0.1, 0.2)	0.605	−0.1 (−0.1, 0.0)	0.036	0.1 (0.01, 0.1)	0.020	0.1 (0.0, 0.1)	0.073	0.2 (0.1, 0.4)	0.006
	Model 2 ^c^ (*n* = 289)	0.1 (0.0, 0.3)	0.060	0.0 (−0.1, 0.2)	0.662	−0.1 (−0.1, 0.0)	0.073	0.1 (0.01, 0.1)	0.046	0.1 (0.0, 0.1)	0.118	0.2 (0.0, 0.4)	0.017
FFM, kg	Model 1 ^d^ (*n* = 352)	0.1 (0.03, 0.2)	0.005	0.0 (−0.1, 0.1)	0.518	0.0 (0.0, 0.05)	0.126	0.0 (−0.1, 0.0)	0.033	−0.1 (−0.1, 0.0)	0.061	−0.1 (−0.2, 0.0)	0.014
	Model 2 ^e^ (*n* = 283)	0.1 (0.01, 0.2)	0.021	0.0 (−0.1, 0.1)	0.600	0.0 (0.01, 0.1)	0.115	0.0 (−0.1, 0.0)	0.031	−0.1 (−0.1, 0.0)	0.033	−0.1 (−0.3, 0.0)	0.013
FM, kg	Model 1 ^d^ (*n* = 352)	0.0 (−0.2, 0.2)	0.706	0.0 (−0.2, 0.2)	0.972	−0.1 (−0.2, 0.0)	0.005	0.2 (0.1, 0.3)	0.001	0.2 (0.1, 0.3)	0.005	0.5 (0.2, 0.7)	<0.001
	Model 2 ^e^ (*n* = 283)	0.0 (−0.2, 0.2)	0.988	0.0 (−0.2, 0.2)	0.878	−0.1 (−0.2, −0.1)	0.027	0.2 (0.0, 0.3)	0.008	0.2 (0.0, 0.3)	0.025	0.4 (0.1, 0.7)	0.012
17 years													
BMI, kg/m^2^	Model 1 ^b^ (*n* = 220)	0.2 (−0.1, 0.5)	0.145	−0.2 (−0.4, 0.1)	0.156	0.0 (−0.1, 0.1)	0.737	0.1 (−0.1, 0.2)	0.317	0.1 (−0.1, 0.2)	0.997	0.3 (−0.1, 0.6)	0.093
	Model 2 ^c^ (*n* = 185)	0.1 (−0.2, 0.4)	0.587	−0.2 (−0.5, 0.1)	0.135	0.0 (−0.1, 0.1)	0.563	0.1 (0.0, 0.2)	0.181	0.1 (−0.1, 0.2)	0.485	0.4 (0.0, 0.8)	0.041
FFM, kg	Model 1 ^d^ (*n* = 205)	0.3 (−0.1, 0.6)	0.119	0.1 (−0.2, 0.4)	0.493	0.1 (−0.1, 0.2)	0.274	−0.1 (−0.3, 0.0)	0.090	0.0 (−0.2, 0.1)	0.617	−0.4 (−0.8, 0.0)	0.037
	Model 2 ^e^ (*n* = 172)	0.3 (−0.1, 0.6)	0.148	0.1 (−0.3, 0.4)	0.751	0.1 (0.0, 0.2)	0.138	−0.2 (−0.3, 0.0)	0.047	−0.1 (−0.3,0.1)	0.312	−0.5 (−0.9, −0.1)	0.029
FM, kg	Model 1 ^d^ (*n* = 205)	0.5 (−0.1, 1.2)	0.114	−0.2 (−0.9, 0.4)	0.454	−0.2 (−0.4, 0.1)	0.240	0.3 (0.0, 0.6)	0.092	0.4 (0.0, 0.8)	0.045	0.9 (0.0, 1.7)	0.041
	Model 2 ^e^ (*n* = 172)	0.2 (−0.6, 0.9)	0.659	−0.2 (−0.9, 0.5)	0.550	−0.2 (−0.5, 0.1)	0.135	0.3 (0.0, 0.6)	0.045	0.4 (0.0, 0.8)	0.067	1.1 (0.2, 2.0)	0.022
9y Serum leptin, ng/mL	Model 1 ^b^ (*n* = 243)	0.0 (−0.5, 0.6)	0.898	−0.1 (−0.6, 0.4)	0.642	−0.1 (−0.3, 0.1)	0.203	0.2 (0.0, 0.5)	0.075	0.2 (−0.1, 0.5)	0.186	0.5 (−0.2, 1.2)	0.127
	Model 2 ^c^ (*n* = 204)	0.1 (−0.3, 0.5)	0.777	0.0 (−0.4, 0.4)	0.927	−0.2 (−0.3, −0.1)	0.036	0.2 (0.1, 0.4)	0.011	0.3 (0.1, 0.6)	0.009	0.7 (0.2, 1.2)	0.008

^a^ Nutrients (% energy) adjusted for total energy according to the multivariate nutrient density model. ^b^ Adjusted for total energy, breastfeeding duration, maternal BMI, maternal education, smoking status, birthweight. ^c^ Model as shown in footnote b + paternal BMI. ^d^ Model shown in footnote b + height. ^e^ Model shown in footnote b + height and paternal BMI; Unadjusted associations at 9 years: Body Mass Index (BMI) (*n* = 406): all *p* values > 0.05 except for: energy (*p* = 0.012), carbohydrate (*p* = 0.019), total fat (*p* = 0.006), MUFA (*p* < 0.001);Fat Free Mass (FFM) (*n* = 396) all *p* values > 0.05 except for: energy (*p* = 0.001); Fat Mass (FM) (*n* = 396): all *p* values > 0.05 except for carbohydrate (*p* = 0.001), total fat (*p* ≤ 0.001), SFA (*p* = 0.007), MUFA ( ≤ 0.001).Unadjusted associations at 17 years: BMI (*n* = 242): all *p* values > 0.05 except for protein (*p* = 0.043), MUFA (*p* = 0.028); FFM (*n* = 226): all *p* values > 0.05 except for energy (*p* = 0.033); FM (*n* = 226) all *p* values > 0.05 except for fat (*p* = 0.047), MUFA (*p* = 0.009). Serum leptin (*n* = 271): unadjusted associations at 9 years all *p* values > 0.05.

**Table 3 nutrients-13-03264-t003:** Multiple linear regression models for energy and energy−adjusted macronutrient intakes of boys (plausible reporters only), using diet collected at 18 months of age to predict body composition, serum leptin concentration at 9 years of age and body composition at 17 years of age in children enrolled in ALSPAC−CiF.

	Model	Energy (0.42 MJ)		Protein (%E) ^a^		Carbohydrate (%E) ^a^		Fat (%E)					
								Total ^a^		SFA ^a^		MUFA ^a^	
		B (95% CI)	*p*-Value	B (95% CI)	*p*-Value	B (95% CI)	*p*-Value	B (95% CI)	*p*-Value	B (95% CI)	*p*-Value	B (95% CI)	*p*-Value
9 years													
BMI, kg/m^2^	Model 1 ^b^ (*n* = 271)	0.5 (0.3, 0.7)	<0.001	0.0 (−0.1, 0.2)	0.928	−0.1 (−0.1, 0.0)	0.122	0.07 (0.0, 0.1)	0.063	0.1(0.0, 0.2)	0.136	0.2 (0.03, 0.4)	0.02
	Model ^c^ (*n* = 216)	0.5 (0.2, 0.7)	<0.001	0.0 (−0.2, 0.1)	0.868	0.0 (−0.1, 0.02)	0.206	0.07 (0.0, 0.1)	0.102	0.1 (0.0, 0.2)	0.221	0.2 (−0.01, 0.4)	0.066
FFM, kg	Model 1 ^d^ (*n* = 264)	0.3 (0.2, 0.4)	<0.001	0.0 (−0.1, 0.1)	0.431	0.0 (0.0, 0.1)	0.622	0.0 (−0.1, 0.0)	0.363	0.0 (−0.1, 0.0)	0.429	−0.1 (−0.2, 0.04)	0.209
	Model 2 ^e^ (*n* = 210)	0.3 (0.2, 0.5)	<0.001	0.0 (−0.1, 0.1)	0.505	0.01 (0.0, 0.1)	0.583	0.0 (−0.1, 0.0)	0.348	0.0 (−0.1, 0.0)	0.319	−0.1 (−0.2, 0.05)	0.210
FM, kg	Model 1 ^d^ (*n* = 264)	0.3 (−0.1, 0.7)	0.115	0.0 (−0.3, 0.2)	0.790	−0.1 (−0.2, 0.0)	0.033	0.2 (0.05, 0.3)	0.006	0.2 (0.0, 0.3)	0.023	0.4 (0.1, 0.7)	0.004
	Model 2 ^e^ (*n* = 210)	0.2 (−0.2, 0.6)	0.314	0.0 (−0.3, 0.2)	0.784	−0.1 (−0.2, 0.0)	0.101	0.1 (0.01, 0.3)	0.032	0.2 (0.0, 0.3)	0.078	0.4 (0.0, 0.7)	0.049
17 years													
BMI, kg/m^2^	Model 1 ^b^ (*n* = 169)	0.3 (−0.1, 0.8)	0.166	−0.1 (−0.5, 0.2)	0.377	0.0 (−0.2, 0.1)	0.611	0.1 (−0.1, 0.2)	0.353	0.1 (−0.1, 0.3)	0.344	0.3 (−0.1, 0.7)	0.160
	Model 2 ^c^ (*n* = 142)	0.2 (−0.3, 0.8)	0.358	−0.2 (−0.6, 0.1)	0.160	0.0 (−0.2, 0.1)	0.715	0.1 (−0.1, 0.3)	0.310	0.1 (−0.1, 0.3)	0.505	0.4 (−0.1, 0.8)	0.104
FFM, kg	Model 1 ^d^ (*n* = 155)	0.5 (0.0, 1.1)	0.060	0.3 (−0.1, 0.7)	0.105	0.0 (−0.1, 0.2)	0.664	−0.1 (−0.3, 0.1)	0.198	0.0 (−0.2, 0.2)	0.858	−0.4 (−0.9., 0.0)	0.068
	Model 2 ^e^ (*n* = 130)	0.7 (0.2, 1.3)	0.015	0.3 (−0.1, 0.7)	0.196	0.1 (−0.1, 0.2)	0.455	−0.1 (−0.3, 0.0)	0.133	−0.1 (−0.3,0.2)	0.582	−0.5 (−1.0, 0.0)	0.055
FM, kg	Model 1 ^d^ (*n* = 155)	0.9 (−0.3, 2.1)	0.136	−0.2 (−1.0, 0.6)	0.620	−0.2 (−0.5, 0.2)	0.318	0.3 (−0.1, 0.6)	0.189	0.4 (0.0, 0.9)	0.075	0.8 (−0.3, 1.8)	0.137
	Model 2 ^e^ (*n* = 130)	0.5 (−0.8, 1.8)	0.447	−0.4 (−1.3, 0.5)	0.412	−0.2 (−0.5, 0.2)	0.397	0.3 (−0.1, 0.7)	0.199	0.4 (−0.1, 0.9)	0.112	0.9 (−0.3, 2.0)	0.133
9y serum leptin, ng/mL	Model 1 ^b^ (*n* = 175)	0.6 (−0.5, 1.6)	0.270	−0.5 (−1.2, 0.1)	0.095	−0.1 (−0.3, 0.2)	0.665	0.2 (−0.1, 0.5)	0.184	0.1 (−0.3, 0.6)	0.533	0.4 (−0.5, 1.2)	0.368
	Model 2 ^c^ (*n* = 148)	−0.2 (−0.9, 0.6)	0.690	−0.3 (−0.7, 0.2)	0.266	−0.1 (−0.3, 0.1)	0.270	0.2 (0.0, 0.4)	0.070	0.3 (0.0, 0.5)	0.078	0.5 (−0.1, 1.0)	0.129

^a^ Nutrients (% energy) adjusted for total energy according to the multivariate nutrient density model. ^b^ Adjusted for total energy, breastfeeding duration, maternal BMI, maternal education, smoking status, birthweight. ^c^ Model as shown in footnote b + paternal BMI. ^d^ Model shown in footnote b + height. ^e^ Model shown in footnote b + height and paternal BMI; Unadjusted associations at 9 years: BMI (*n* = 301): all *p* values > 0.05 except for: energy (*p* ≤ 0.001), total fat (*p* = 0.014), MUFA (*p* = 0.001); FFM (*n* = 292) all *p* values > 0.05 except for: energy (*p* < 0.001); FM (*n* = 292): all *p* values > 0.05 except for carbohydrate (*p* = 0.014), total fat (*p* = 0.001), SFA (*p* = 0.019) and MUFA (*p* ≤ 0.001). Unadjusted associations at 17 years: BMI (*n* = 184): all *p* values > 0.05 except for MUFA (*p* = 0.048); FFM (*n* = 170): all *p* values > 0.05 except for energy (*p* = 0.012); FM (*n* = 170) all *p* values > 0.05 except for MUFA (*p* = 0.039). Serum leptin (*n* = 271): unadjusted associations at 9 years all *p* values > 0.05.

**Table 4 nutrients-13-03264-t004:** Multiple linear regression models for energy and energy−adjusted macronutrient intakes of girls, using diet collected at 18 months of age to predict body composition and serum leptin concentration at 9 years of age and body composition at 17 years of age in children enrolled in ALSPAC−CiF.

	Model	Energy/0.42 MJ		Protein %E ^a^		Carbohydrate (%E) ^a^		Fat (%E)					
								Total ^a^		SFA ^a^		MUFA ^a^	
		B (95% CI)	*p*-Value	B (95% CI)	*p*-Value	B (95% CI)	*p*-Value	B (95% CI)	*p*-Value	B (95% CI)	*p*-Value	B (95% CI)	*p*-Value
9 years													
BMI, kg/m^2^	Model 1 ^b^ (*n* = 308)	0.0 (−0.1, 0.1)	0.970	0.0 (−0.2, 0.1)	0.725	0.0 (−0.03, 0.1)	0.347	0.0 (−0.1, 0.04)	0.348	−0.03 (−0.1,0.1)	0.548	0.0 (−0.2, 0.2)	0.872
	Model 2 ^c^ (*n* = 246)	0.0 (−0.2, 0.1)	0.927	−0.1 (−0.2, 0.1)	0.273	0.0 (−0.02, 0.1)	0.258	0.0 (−0.1, 0.04)	0.412	0.0 (−0.1, 0.04)	0.337	0.1 (−0.1, 0.3)	0.537
FFM, kg	Model 1 ^e^ (*n* = 302)	0.0 (−0.1, 0.1)	0.430	0.0 (−0.1, 0.1)	0.985	0.0 (−0.02, 0.05)	0.425	0.0 (−0.1, 0.0)	0.318	0.0 (−0.1, 0.0)	0.223	0.0 (−0.1, 0.1)	0.812
	Model 2 ^f^ (*n* = 241)	0.0 (−0.1, 0.1)	0.472	0.0 (−0.1, 0.1)	0.632	0.0 (−0.02, 0.06)	0.320	0.0 (−0.1, 0.0)	0.330	0.0 (−0.1, 0.2)	0.210	0.0 (−0.2, 0.1)	0.742
FM, kg	Model 1 ^e^ (*n* = 302)	0.0 (−0.2, 0.2)	0.968	−0.1 (−0.3, 0.2)	0.571	0.00 (−0.1, 0.1)	0.955	0.0 (−0.1, 0.1)	0.813	0.0 (−0.1, 0.2)	0.651	0.2 (−0.1, 0.5)	0.226
	Model 2 ^f^ (*n* = 241)	0.0 (−0.2, 0.2)	0.869	−0.1 (−0.3, 0.2)	0.549	−0.02 (−0.1, 0.1)	0.630	0.1 (−0.1, 0.2)	0.369	0.03 (−0.1, 0.2)	0.640	0.3 (0.0, 0.6)	0.031
17 years													
BMI, kg/m^2^	Model 1 ^b^ (*n* = 229)	−0.1 (−0.4, 0.2)	0.462	−0.05 (−0.3, 0.2)	0.639	0.00 (−0.1, 0.1)	0.983	0.0 (−0.1, 0.1)	0.780	0.0 (−0.1, 0.2)	0.769	0.1 (−0.2, 0.4)	0.547
	Model 2 ^c^ (*n* = 184)	−0.1 (−0.4, 0.2)	0.609	−0.1 (−0.4, 0.1)	0.271	0.02 (−0.1, 0.1)	0.725	0.0 (−0.1, 0.1)	0.897	0.0 (−0.2, 0.1)	0.753	0.2 (−0.1, 0.5)	0.240
FFM, kg	Model 1 ^d^ (*n* = 214)	−0.2 (−0.4, 0.0)	0.109	−0.1 (−0.3, 0.04)	0.125	0.1 (0.0, 0.2)	0.064	−0.1 (−0.2, 0.0)	0.154	−0.1 (−0.2, 0.0)	0.048	−0.1 (−0.3, 0.2)	0.502
	Model 2 ^e^ (*n* = 172)	−0.2 (−0.5, 0.0)	0.084	−0.2 (−0.4, 0.0)	0.057	0.1 (0.0, 0.2)	0.018	−0.1 (−0.2, 0.0)	0.059	−0.2 (−0.3, −0.0)	0.012	−0.1 (−0.4, 0.2)	0.481
FM, kg	Model 1 ^d^ (*n* = 214)	−0.2 (−0.8, 0.4)	0.475	−0.1 (−0.5, 0.4)	0.732	−0.01 (−0.2, 0.2)	0.942	0.0 (−0.2, 0.3)	0.773	0.1 (−0.2, 0.4)	0.607	0.1 (−0.6, 0.7)	0.798
	Model 2 ^e^ (*n* = 172)	−0.1 (−0.7, 0.5)	0.679	−0.2 (−0.7, 0.3)	0.341	0.03 (−0.2, 0.2)	0.817	0.0 (−0.2, 0.3)	0.831	0.0 (−0.3, 0.3)	0.987	0.4 (−0.3, 1.1)	0.302
9y serum leptin, ng/mL	Model 1 ^b^ (*n* = 200)	−0.3 (−0.9, 0.3)	0.387	0.0 (−0.6, 0.5)	0.938	0.1 (−0.1, 0.4)	0.368	−0.2 (−0.5, 0.1)	0.280	−0.2 (−0.6 0.2)	0.366	−0.1 (−1.0, 0.7)	0.740
	Model 2 ^c^ (*n* = 161)	−0.2 (−0.9, 0.4)	0.474	−0.4 (−1.0, 0.2)	0.192	0.3 (0.0, 0.5)	0.055	−0.3 (−0.6, 0.0)	0.086	−0.4 (−0.8, 0.0)	0.061	−0.2 (−1.1, 0.7)	0.659

^a^ Nutrients (% energy) adjusted for total energy according to the multivariate nutrient density model. ^b^ Adjusted for total energy, breastfeeding duration, maternal BMI, maternal education, smoking status, birthweight. ^c^ Model as shown in footnote b + paternal BMI. ^d^ Model shown in footnote b + height. ^e^ model shown in footnote b + height and paternal BMI; Unadjusted associations at 9 years: BMI (*n* = 345): all *p* values > 0.05; FFM (*n* = 336) all *p* values > 0.05; FM (*n* = 336): all *p* values > 0.05 Unadjusted associations at 17 years: BMI (*n* = 258): all *p* values > 0.05; FFM (*n* = 242): all *p* values > 0.05; FM (*n* = 242) all *p* values > 0.05. Serum leptin (*n* = 223): unadjusted associations at 9 years all *p* values > 0.05.

**Table 5 nutrients-13-03264-t005:** Multiple linear regression models for energy and energy—adjusted macronutrient intakes of girls (plausible reporters only), using diet collected at 18 months of age to predict body composition, serum leptin concentration at 9 years of age and body composition at 17 years of age in children enrolled in ALSPAC-CiF.

	Model	Energy/0.42 MJ		Protein (%E) ^a^		Carbohydrate (%E) ^a^		Fat (%E)					
								Total ^a^		SFA ^a^		MUFA ^a^	
		B (95% CI)	*p*-Value	B (95% CI)	*p*-Value	B (95% CI)	*p*-Value	B (95% CI)	*p*-Value	B (95% CI)	*p*-Value	B (95% CI)	*p*-Value
9 years													
BMI, kg/m^2^	Model 1 ^b^ (*n* = 216)	0.3 (0.1, 0.6)	0.002	−0.1 (−0.2, 0.1)	0.416	0.0 (0.0, 0.1)	0.466	0.0 (−0.1, 0.1)	0.678	0.0 (−0.1, 0.1)	0.662	0.1 (−0.1, 0.3)	0.434
	Model 2 ^c^ (*n* = 173)	0.3 (0.1, 0.6)	0.017	−0.1 (−0.2, 0.1)	0.352	0.0 (−0.1, 0.1)	0.813	0.0 (−0.1, 0.1)	0.839	0.0 (−0.1, 0.1)	0.768	0.1 (−0.1, 0.3)	0.350
FFM, kg	Model 1 ^d^ (*n* = 211)	0.0 (−0.1, 0.2)	0.667	0.0 (−0.1, 0.1)	0.748	0.0 (−0.04, 0.04)	0.881	0.0 (−0.1, 0.04)	0.714	0.0 (−0.1, 0.04)	0.536	0.0 (−0.1, 0.2)	0.562
	Model 2 ^e^ (*n* = 169)	0.0 (−0.2, 0.2)	0.808	0.0 (−0.1, 0.1)	0.598	0.0 (−0.04, 0.05)	0.883	0.0 (−0.1, 0.1)	0.928	0.0 (−0.1, 0.1)	0.764	0.0 (−0.1, 0.2)	0.716
FM, kg	Model 1 ^d^ (*n* = 211)	0.4 (0.1, 0.8)	0.019	−0.2 (−0.4, 0.0)	0.099	0.0 (−0.1, 0.1)	0.659	0.0 (−0.1, 0.1)	0.716	0.0 (−0.1, 0.2)	0.631	0.2 (−0.1, 0.5)	0.160
	Model 2 ^e^ (*n* = 169)	0.5 (0.1, 0.9)	0.024	−0.1 (−0.3, 0.1)	0.419	0.0 (−0.1, 0.1)	0.479	0.1 (0.0, 0.2)	0.181	0.1 (−0.1, 0.3)	0.160	0.3 (0.0, 0.7)	0.049
17 years													
BMI, kg/m^2^	Model 1 ^b^ (*n* = 168)	0.3 (−0.1, 0.7)	0.091	0.0 (−0.2, 0.3)	0.720	0.0 (−0.1, 0.1)	0.586	0.0 (−0.1, 0.2)	0.613	0.1 (−0.1, 0.2)	0.367	0.1 (−0.2, 0.4)	0.573
	Model 2 ^c^ (*n* = 135)	0.3 (−0.1, 0.7)	0.190	−0.1 (−0.3, 0.2)	0.619	0.0 (−0.1, 0.1)	0.493	0.1 (−0.1, 0.2)	0.275	0.1 (−0.1, 0.2)	0.303	0.3 (−0.1, 0.6)	0.110
FFM, kg	Model 1 ^d^ (*n* = 159)	0.0 (−0.3, 0.3)	0.922	0.0 (−0.2, 0.2)	0.877	0.04 (0.0, 0.1)	0.328	−0.1 (−0.2, 0.0)	0.262	−1.0 (−0.2, 0.0)	0.118	0.0 (−0.3, 0.3)	0.985
	Model 2 ^e^ (*n* = 128)	−0.1 (−0.5, 0.3)	0.545	−0.2 (−0.4, 0.1)	0.188	0.1 (0.0, 0.2)	0.173	−0.1 (−0.2, 0.1)	0.309	−0.1 (−0.3, 0.0)	0.150	0.0 (−0.3, 0.3)	0.867
FM, kg	Model 1 ^d^ (*n* = 159)	0.7 (−0.2, 1.5)	0.124	0.2 (−0.3, 0.7)	0.503	−0.1 (−0.3, 0.2)	0.674	0.0 (−0.2, 0.3)	0.825	0.2 (−0.2, 0.5)	0.394	0.0 (−0.7, 0.8)	0.945
	Model 2 ^e^ (*n* = 128)	0.7 (−0.2, 1.6)	0.144	−0.1 (−0.6, 0.5)	0.834	−0.1 (−0.3, 0.2)	0.651	0.1 (−0.2, 0.4)	0.485	0.1 (−0.2, 0.5)	0.479	0.5 (−0.2, 1.3)	0.174
9y serum leptin, ng/mL	Model 1 ^b^ (*n* = 143)	0.2 (−0.9, 1.3)	0.692	−0.4 (−1.1, 0.3)	0.257	0.1 (−0.2, 0.4)	0.360	−0.1 (−0.5, 0.3)	0.611	−0.1 (−0.6, 0.4)	0.665	0.1 (−0.9, 1.1)	0.877
	Model 2 ^c^ (*n* = 118)	0.1 (−1.2, 1.3)	0.904	−0.6 (−1.4, 0.2)	0.116	0.2 (−0.1, 0.5)	0.254	−0.1 (−0.5, 0.3)	0.561	−0.1 (−0.6, 0.4)	0.729	0.0 (−1.1, 1.1)	0.980

^a^ Nutrients (% energy) adjusted for total energy according to the multivariate nutrient density model. ^b^ Adjusted for total energy, breastfeeding duration, maternal BMI, maternal education, smoking status, birthweight. ^c^ Model as shown in footnote b + paternal BMI. ^d^ Model shown in footnote b + height. ^e^ Model shown in footnote b + height and paternal BMI; Unadjusted associations at 9 years: BMI (*n* = 238): all *p* values > 0.05 except energy (*p* = 0.010); FFM (*n* = 231) all *p* values > 0.05; FM (*n* = 231): all *p* values > 0.05 except energy (*p* = 0.042). Unadjusted associations at 17 years: BMI (*n* = 186): all *p* values > 0.05 except SFA (*p* = 0.017); FFM (*n* = 176): all *p* values > 0.05; FM (*n* = 176) all *p* values > 0.05 except SFA (0.033). Serum leptin (*n* = 160): unadjusted associations at 9 years all *p* values > 0.05.

**Table 6 nutrients-13-03264-t006:** Multiple linear regression models for energy—adjusted fat intakes of boys and girls (classified into groups according to international guidelines [18]), using diet collected at 18 months of age to predict body composition at 9 and 17 years of age in children enrolled in ALSPAC-CiF.

Fat Intake % Energy	Fully Adjusted Model ^a^				
	*n*	Mean BMI (kg/m^2^)	B	95% CI	*p* Value
Boys					
BMI 9 years					
Group 1 <35%	86	17.0	0		
Group 2 35–40%	132	17.6	0.6	−0.1, 1.3	0.028
Group 3 >40%	71	18.1	1.1	0.3, 1.9	
Fat mass 9 years					
Group 1 <35%	86	6.5	0		0.003
Group 2 35–40%	127	7.8	1.3	0.1, 2.4	
Group 3 >40%	70	8.8	2.3	0.9, 3.5	
BMI 17 years					
Group1 <35%	52	22.6	0		
Group2 35–40%	83	24.1	1.5	0.1, 2.8	0.048
Group3 >40%	50	24.4	1.8	0.2, 3.3	
Fat mass 17 years					
Group 1 <35%	47	11.3	0		0.009
Group 2 35–40%	77	16.0	4.7	1.3, 8.1	
Group 3 >40%	48	16.4	5.1	1.4, 8.9	
Girls					
BMI 9 years					
Group 1 <35%	64	18.2	0		
Group 2 35–40%	117	17.4	−0.8	−1.5, 0.2	0.143
Group 3 >40%	65	17.5	−0.7	−1.5, 0.0	
Fat mass 9 years					
Group 1 <35%	62	9.5	0		0.534
Group 2 35–40%	115	9.0	−0.5	−1.8, 0.6	
Group 3 >40%	64	9.5	0	−1.4, 1.3	
BMI 17 years					
Group 1 <35%	49	23.2	0		
Group 2 35–40%	90	21.7	−1.5	−2.9, −0.2	0.042
Group 3 >40%	45	22.8	−0.4	−1.8, 1.2	
Fat mass 17 years					
Group 1 <35%	46	22.7	0		0.109
Group 2 35–40%	86	19.8	−2.9	−4.4, 2.6	
Group 3 >40%	40	21.8	−0.9	−5.8, −0.003	

^a^ Adjusted for total energy, breastfeeding status, maternal education, maternal BMI, smoking in pregnancy, paternal BMI.

## Data Availability

The data presented in this study are available on request from the ALSPAC. The data are not publicly available as there are charges for using ALSPAC data.

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
