# Peer review of "Association of Nutrition in Early Childhood with Body Composition and Leptin in Later Childhood and Early Adulthood"

_nutrients, 2021, doi:10.3390/nu13093264_

Round 1
Reviewer 1 Report
The authors of this manuscript aimed to replicate the analyses done previously on the ELANCE cohort, where the impact of early nutrition on adult obesity was studied. Additionally authors wanted to expand their analyses by incorporating more confounding variables in their analyses and also by studying those association in the subgroups divided by sex.
The study seems to be interesting, since this kind of data may add new insights into reasons of obesity development in adults. It can also raises awareness of people by showing that early nutrition may be crucial for health in later life.
The reasoning for doing this project showed by the authors seems reasonable. There is only one point in this reasoning that I wouldn’t agree with – the authors wrote that in the ELANCE cohort the variance in fat intake was not big and was generally low, however they refer to the mean value, which does not relate to the distribution of fat intake.
In my opinion the impact of other confounders that are well known contributors to obesity were not taken into account: physical activity and current dietary intake. For example in models investigating impact of %energy from fat on BMI or FM/FFM also the % of fat intake of 20 years old adults should be taken into account (as well as physical activity). As far I see, the data concerning physical activity are not available, but the data concerning the intake are available. This kind of analysis can show whether the impact of early nutrition is greater than the impact of current nutrition. Additionally reporting adjusted R2 for all models would add a lot of value to the paper.
What is also worth mentioning is that the intake was evaluated only from 3 days, which may not reflect long term dietary habits. In case of small children (8 months) they do not vary so much from day to day, but in case of older children and adults this may vary a lot from day to day. So at least it should be discussed in the discussion part describing limitations.
Additional comments are described below in the order of appearance:
- There is no conclusion at the end of the abstract
- An information should be provided regarding the time of blood drawing since there is diurnal variation in leptin concentration in blood
- Lines 166-169 – in my opinion the comparison between people that dropped out from the study with the people that stayed in the study until the end. Additionally supplementary table 1 lacks many details, like: units, number of participants in each column, whether mean or median is presented, no parameters showing spread of data points are shown (sd or iqr)
- The study flow and indication of time points for drop outs (and the number) would be very informative
- Table 1 – there is no information about the number of boys and girls at the age of 9 and 17 years
- In titles of S2 and S3 there should be references to ELANCE study instead of giving the name of first author of this paper
- Conclusions resemble part of an abstract. Instead, some generalizations should be made based on the results that were found.
- In the discussion more information should be given on the result concerning leptin concentrations, since this is the part underlined in the title of manuscript. Additionally an information about the meaning of the differences in the obtained result between using the intake from 8 and 18 months old children. Does it imply some possible recommendation for diet of those children in this age? Does it show that incorporating new food products may be more critical time period than first months (according to obesity in later life)?
Reviewer 2 Report
Your research is interesting and I consider that the results you have obtained are relevant.
Statistical analyzes must include the p value if it is <0.05 and indicate which is not significant (p>0.05) without specifying the value.
The article has too many bibliographic references. You should try to limit the number to 35.
Author Response
Thank you for this comment
We prefer to give the exact p value for associations wherever possible – this allows the reader to make a judgement of the strength or weakness of the associations without reference to the arbitrary cut off of 0.05.
We have included less than 50 reference citations, which we do not consider excessive for a research article. We have complied with journal guidelines on this
Reviewer 3 Report
The study is partly a replication of previous prospective work on associations between nutrition in early childhood and body composition and leptin in later childhood and adolescence, but is particularly useful in its inclusion of additional data and larger sample sizes allowing to study differences between sexes.
I have several questions and concerns with this manuscript:
The abstract is succinct but inclusion of numerical values in support of the statements would be helpful. In addition, acronyms should be defined.
The introduction and discussion would benefit from an updated literature review, most of the cited papers were older than 10 years.
Is it possible to distinguish different carbohydrates in ALSPAC data? It would be interesting to also analyse the associations with complex carbohydrates or fibre and sugars separately, as was done for different types of fats.
How was the amount of breast milk assessed by parents and was its changing macronutrient composition considered while calculating total energy intake?
Would it be possible to adjust results also to nutrition (macronutrient intakes or dietary patterns) at age of 9 and 17 years?
Breastfeeding was categorised as never breastfed\< 1 month vs ≥1 month. Was effect of breastfeeding for longer period considered? If there is no additional data, then data collected at 8 and 18 months of age (22,6% and 2,3%, respectively) can be used. Did formula milk and cow’s milk have any effect on the results?
The study should be restricted to participants who have complete data on all timepoints. The various group sizes are confusing and do not allow comparisons, nor do they provide an adequate overall view.
It is unclear whether the children with fat %E <30% were excluded or were they included in the group of %E <35%. Would exclusion of them affect the results?
Table S1 lacks group sizes (n). The units are also missing (weight and length at birth, BMI, energy intakes, gestation, mother age).
Tables S2-S3 – citing of ELANCE study in the title should be formalized.
Table 1 – the group sizes of 9 and 17 years are missing.
Round 2
Reviewer 1 Report
Concerning authors’ response I still have some doubts, which I list below:
- Please give information in the text about the intake of fat (as percentage of energy) with the accuracy to decimal place (as it was given in Rolland-Cachera,et al and as You give in Your results), so it is possible to compare the data in both cohorts
- I agree that confounder needs to influence both dependent and independent variables and thus the reasoning for not including those variables in the model as a confounders is clear for me. However I still think that showing the relation between early and adult nutrition and between adult nutrition and body fat would be informative. In fact not showing it, may lead to false conclusions (for example if a child ate a lot of fat and still eats a lot of fat as an adult, which of those facts contribute to body fat? It is not possible to separate those and then to draw proper conclusions)
- I cannot see an answer concerning adding R2 to the models (and also R2 was not added to the tables)
- I still think that at least 2 first sentences of conclusions provided in the abstract are not conclusions but summary. Additionally please include in the conclusion what parameters of the diet (eg. high fat intake) affects high body fat, so that reader knows immediately what is the direction of association.
- I asked to indicate when the blood was drawn for leptin determination. Authors added “during daytime clinics”. I suppose it should be “during daytime in clinics”?? Moreover, this does not fully answer my suggestion – I would at least expect a time range and also some comment on this in the discussion section, since I see that the concentration of leptin could have varied not only because of BMI but also because of the timing of blood drawing.
- In my opinion the reason for showing the data in S1 should be explained as it does not arise from aims.
- Discussion concerning leptin – in my opinion there would be something to add here. For example if early nutrition does not impact leptin, than what has the impact? What other studies showed? What about current nutrition?
Author Response
file included

Reviewer 3 Report
There is no precision "0.0" or "0.00", please change either to 0 or <0.01 (Abstract, lines 64 and 67).
The Authors have nicely answered to the question about breast milk intake assessment, I suggest to include this information also in the section of Methods.
Author Response
see file attaced
